# Heterodyne analysis of high-order partial waves in attosecond photoionization of helium

Wenyu Jiang[1,8,9], Luke Roantree [2,9], Lulu Han[1,9], Jiabao Ji [3], Yidan Xu[1], Zitan Zuo[1], Hans Jakob Wörner[3], Kiyoshi Ueda [1,4,5], Andrew C. Brown [2] ✉, Hugo W. van der Hart [2], Xiaochun Gong[1] & Jian Wu [1,6,7] ✉

Partial wave analysis is key to interpretation of the photoionization of atoms and molecules on the attosecond timescale. Here we propose a heterodyne analysis approach, based on the delay-resolved anisotropy parameters to reveal the role played by high-order partial waves during photoionization. This extends the Reconstruction of Attosecond Beating By Interference of Two-photon Transitions technique into the few-photon regime. We demonstrate that even for moderate ( ~ 1TW/cm$^2$) intensities, near-infrared-assisted photoionization of helium through Rydberg states results in a tiny contribution from the $g_0$ wave, which has a significant impact on the photoelectron angular distributions via interference with the $s$- and $d_0$-waves. This modulation also causes a substantial deviation in the angular distribution of the recovered spectral phase shift. Our analysis provides an efficient method to resolve isolated partial wave contributions beyond the perturbative regime, and paves the way towards understanding resonance-enhancement of partial waves.

By characterizing photoionization as a half-scattering process[1], the continuum photoelectron wave-packet may be decomposed into a set of angular- and magnetic-quantum-number-resolved channels. This 'partial wave analysis' provides an effective method for interpreting experimental observations, especially for comprehending the mechanism of complex resonances[2–4], interpreting the origin of the angular dependence of photoionization time-delays[5–9] and resolving the quantum transition interference[10]. In particular, it is the interference between partial waves which is encoded within photoelectron angular distributions (PADs)[11]. Thus the precise detection and characterization of individual partial wave contributions– and their interference– is key to revealing the scattering-channel-resolved mechanisms of light-matter interactions.

The state of the art in attosecond-metrology is the measurement of photoelectron emission time delays which encode information about the atomic potential[12–18] or molecular structure[19,20] through scattering of the outgoing photoelectron by the atomic or molecular potential. One of the most commonly used methods to determine ionization time delays is the reconstruction of attosecond beating by interference of two-photon transitions (RABBITT) method, which employs a near-infrared (NIR) pulse and a time-delayed extreme-ultraviolet attosecond pulse train (XUV-APT) to drive two-photon ionization[21]. In a general experiment, RABBITT employs a NIR pulse with an intensity on the order of $10^{10}$ to $10^{11}$ W/cm$^2$ in atoms, and slightly higher in molecules and clusters due to their lower cross sections[22–24]. This limits the number of photon absorptions/emissions

[1]State Key Laboratory of Precision Spectroscopy, East China Normal University, Shanghai, China. [2]Centre for Light-Matter Interaction, School of Mathematics and Physics, Queen's University Belfast, Northern Ireland, United Kingdom. [3]Laboratorium für Physikalische Chemie, ETH Zürich, Zürich, Switzerland. [4]Department of Chemistry, Tohoku University, Sendai, Japan. [5]School Physical Science and Technology, ShanghaiTech University, Shanghai, China. [6]Collaborative Innovation Center of Extreme Optics, Shanxi University, Taiyuan, Shanxi, China. [7]Chongqing Key Laboratory of Precision Optics, Chongqing Institute of East China Normal University, Chongqing, China. [8]Present address: School of Physics and Microelectronics, Zhengzhou University, Zhengzhou, China. [9]These authors contributed equally: Wenyu Jiang, Luke Roantree, Lulu Han. ✉e-mail: andrew.brown@qub.ac.uk; jwu@phy.ecnu.edu.cn

which may contribute to the RABBITT spectrum to two– one NIR and one XUV. Only partial waves that follow the two-photon transition rules contribute to the observed sidebands[8,25], but at higher intensities, higher-order partial waves arise through multi-photon transitions. In these cases, the overall signal remains dominated by these one- or two-photon transitions, making it extremely difficult to isolate the contributions of the higher-order partial waves.

In this work, we outline a technique akin to heterodyne detection, using the interference between partial waves to characterize the high-order contribution to RABBITT in helium atoms. Our experimental observations, supported by theoretical simulations using the R-matrix with time-dependence (RMT) code[26–28], demonstrate that even a fractional contribution from higher-order processes can significantly modify the photoionization time delay through partial wave interference.

## Results

As illustrated in Fig. 1a, the experimental measurements were performed using an attosecond coincidence interferometer– based on a near-infrared (NIR) 775 nm femtosecond laser pulse– with a high momentum resolution for photoelectron kinetic energy below 7 eV. The photoionization-induced three-dimensional momentum of ion fragments and photoelectrons was measured in coincidence using a cold target recoil ion momentum spectroscopy setup[29,30]. The ion fragments and photoelectrons were guided by a homogeneous electric and magnetic field towards their respective detectors. The XUV-APT was generated via high harmonic (HH) generation from krypton atoms, covering XUV photon energies from 20.8 eV (HH13) to 30.4 eV (HH19), and a co-propagated, 532 nm reference laser was used to actively stabilize the relative phase between XUV-APT and NIR field with a time-jitter below 30 as. Figure 1b shows a photoionization pathway diagram starting from the He $1s^2$ ground state in the presence of the parallel,

linearly polarized XUV-APT and NIR field with an intensity of 1 TW/cm². The standard two-photon RABBITT process is depicted on the left of Fig. 1b. Absorption of an XUV photon from the APT results in ionization, with the photoelectron attaining an energy in one of several so-called "main bands" (MB). Absorption or emission of a NIR photon then leaves the photoelectron in a "sideband" (SB), in between two main bands. The two possible pathways to a given SB (say SB16) are absorption of HH15 plus a NIR photon, or absorption of HH17 and emission of a NIR photon. The interference between these two pathways is made manifest by varying the time delay between the XUV-APT and NIR pulse. The SB yield then oscillates with a frequency of twice the NIR photon energy ($2\omega_{NIR}$), and the phase shift of the oscillation can be used to reveal the photoionization dynamics.

At higher NIR intensities, besides these standard, two-photon RABBITT pathways, additional four-photon pathways may also contribute to the generation of the SB($2q$) photoelectrons. The additional four-photon pathways reaching SB($2q$) are

1. absorption of a HH($2q − 1$) or HH($2q + 1$) photon, followed by two NIR absorptions (emissions) and one NIR emission (absorption),
2. absorption of a HH($2q − 3$) photon and three NIR photons,
3. absorption of a HH($2q + 3$) photon and emission of three NIR photons.

The selection rules for this four-photon ionization are $\Delta l = 0, \pm 2, \pm 4$ and $\Delta m = 0$, meaning the SB results from a coherent superposition of $s$-, $d_0$- and $g_0$-waves in the helium case. Higher-order processes will not be visible in the angle-integrated SB yield, but, as we shall see, PADs can contain signatures of the $g_0$ and $i_0$-wave via interference between different partial waves.

Figure 2 b and g display the RABBITT spectra for a NIR intensity of 1.1 TW/cm² (experiment) and 1.0 TW/cm² (theory), and Fig. 2a and f are

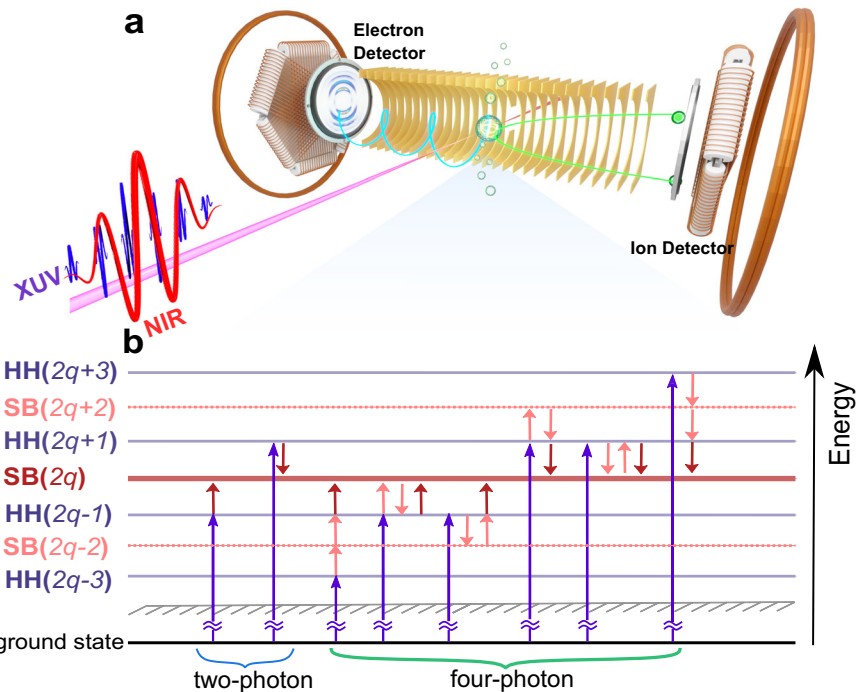

**Fig. 1 | Schematic of the attosecond coincidence interferometer. a** Experimental configuration. **b** Transition map of photoionization in helium atoms. In the two-photon regime shown by blue brace, the interference between HH($2q − 1$) and HH($2q + 1$) pathways via absorption and emission of one NIR photon leads to a $2\omega_{NIR}$ oscillation in the yields of SB($2q$). In the four-photon regime shown by green brace, the interference between (HH($2q − 3$)+3NIR) and (HH($2q − 1$)+NIR) pathways,

(HH($2q − 1$)+NIR) and (HH($2q + 1$)-NIR) pathways, (HH($2q + 1$)-NIR) and (HH($2q + 3$)-3NIR) pathways also contributes to the $2\omega_{NIR}$ oscillations, where the plus (minus) symbol represents the absorption (emission) of NIR photons, and $\omega_{NIR}$ is the angular frequency of the fundamental NIR field. Additional $4\omega_{NIR}$ oscillations arise from the four-photon-transition-induced interference between (HH($2q − 3$)+3NIR) and (HH($2q + 1$)-NIR), or (HH($2q − 1$)+NIR) and (HH($2q + 3$)-3NIR) pathways.

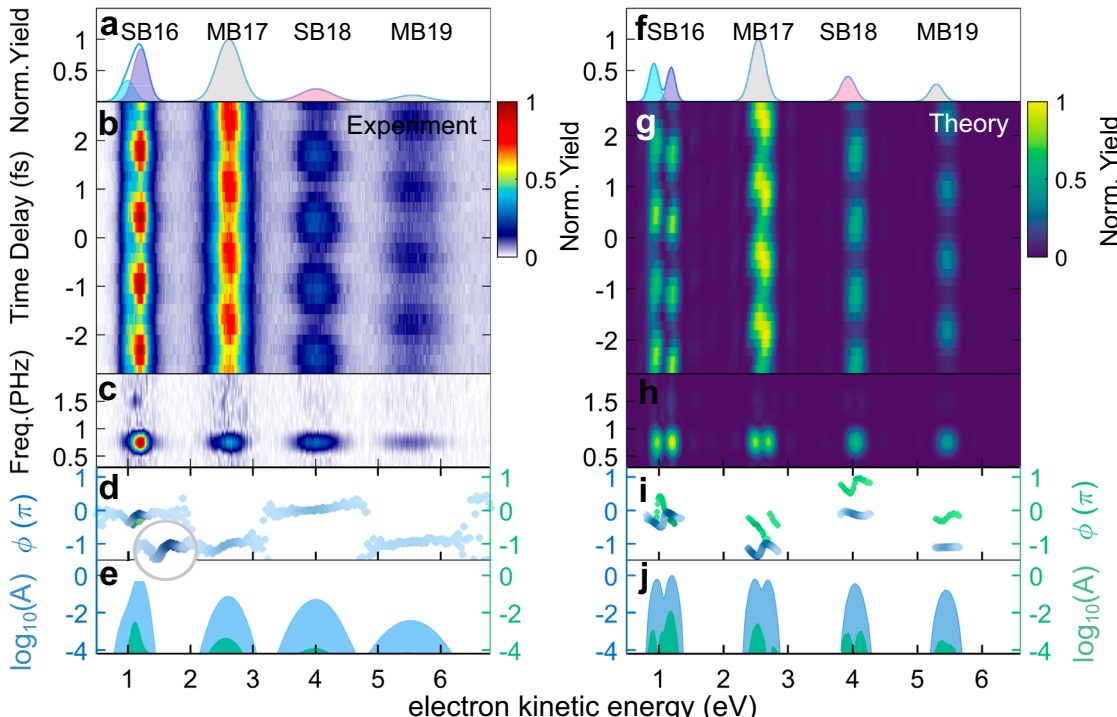

**Fig. 2 | Attosecond photoelectron kinetic energy spectrum. a** Experimentally measured photoelectron kinetic energy spectrum. Shadows represent the results of multi-peak Gaussian fitting, where the blue, purple and red ones represent the fitted results of SB16(4p), SB16(5p) and SB18, and the gray ones are results of MB17 and MB19. **b** Attosecond-resolved photoelectron spectrum with a NIR intensity of $1.1 \times 10^{12}$ W/cm². **c, d** FFT amplitude spectrum and phase shift distributions. In (**d**), blue and the green dots represent phase shifts of $2\omega_{\text{NIR}}$ and $4\omega_{\text{NIR}}$ oscillations, respectively. **e** The gated amplitude distributions at $2\omega_{\text{NIR}}$ (blue area) and $4\omega_{\text{NIR}}$ (green area) on a base-10 logarithmic scale. **f–j** As (**a–e**), but for RMT simulations with a NIR intensity of $1.0 \times 10^{12}$ W/cm².

the corresponding photoelectron kinetic energy spectra. The theoretical results (Fig. 2g) are calculated using the R-matrix with time-dependence (RMT) code[26–28] based on a chirp-less XUV-APT spanning HH13 to HH21 of the NIR pulse with a central wavelength of 775.5 nm. Figure 2c and h present the corresponding fast Fourier transform (FFT) amplitudes. The appearance of a $4\omega_{\text{NIR}}$ oscillation in addition to $2\omega_{\text{NIR}}$ in the first sideband indicates that the four-photon transitions become non-negligible for TW-level NIR intensities.

The near-threshold SB16 in helium results from the interference between a laser-assisted resonant ionization pathway and direct photoionization pathway[31–33]. The absorption of one HH15-photon can simultaneously excite the $1s4p$ and $1s5p$ Rydberg states[15,34–39]. Subsequent absorption of a NIR photon leads to ionization into SB16, which may also be reached by absorption of HH17 and emission of a NIR photon. The interference between the direct and resonant pathways results in two discrete peaks in SB16 due to an excitation energy gap of 0.3 eV. This double peak structure is visible in the theoretical results in Fig. 2g but it is obscured by the relatively lower kinetic energy resolution in the experimental spectrum in Fig. 2b. However, the interference also modifies the phase of the resulting $2\omega_{\text{NIR}}$ oscillation, revealing the signature of photoionization from the double Rydberg states which can be seen both in the inset in Fig. 2d and in Fig. 2i[37,40,41]. For the resonant-states-imprinted SB16, both $\phi^{2\omega}_{\text{SB16}(4p)}$ and $\phi^{2\omega}_{\text{SB16}(5p)}$ in Fig. 2d and i display a negative slope as a function of kinetic energy, which is different from the flat, two-photon phase shift of SB18[31].

At this NIR intensity, the SB yields oscillate as $Y_{\text{SB}} = A\cos(2\omega_{\text{NIR}}\tau - \phi^{2\omega}_{\text{SB}}) + B\cos(4\omega_{\text{NIR}}\tau - \phi^{4\omega}_{\text{SB}}) + C$. Due to the additional 4-photon transitions, both the $2\omega_{\text{NIR}}$ and $4\omega_{\text{NIR}}$ yields comprise multiple 'component' signals from different interfering ionization pathways. The interference between HH($2q + 1$) and HH($2q - 1$) pathways from both the two- and four-photon transitions contribute to the $2\omega_{\text{NIR}}$ term, which has a total yield of A. Most notably, the $4\omega_{\text{NIR}}$ yield

contains contributions from a) interference between ionization paths (HH($2q - 1$) + 1) and (HH($2q + 3$) − 3), and b) interference between (HH($2q - 3$) + 3) and (HH($2q + 1$) − 1) where the plus (minus) symbol represents absorption (emission) of NIR photons. These component signals have identical periods but differ in phase. Up to the second order in the phase-difference, the total phase (parameter $\phi^{4\omega}_{\text{SB}}$) is given by the weighted average of the component phases, and the total yield (parameter B) reflects the sum of the component yields, see Supplementary Note 4 in the Supplementary Information (SI). The non-oscillating term, parameter C represents the delay-averaged yield in the sideband.

The phase shift in the $2\omega_{\text{NIR}}$ term is defined as $\phi^{2\omega}_{\text{NIR}} = \Delta\phi_{\text{Wig.}} + \Delta\phi_{\text{cc}} + \Delta\phi_{\text{XUV}}$, where $\Delta\phi_{\text{XUV}}$ is the 'Atto-chirp'[42], $\Delta\phi_{\text{Wig.}}$ is the one-photon scattering phase shift[43,44], and $\Delta\phi_{\text{cc}}$ is the continuum-continuum phase shift due to the NIR field[45]. The phase of the $4\omega_{\text{NIR}}$ oscillation includes two degenerate components: $\Delta\phi^{(2q-3,\,2q+1)}_{\text{Wig.}} + \Delta\phi^{(2q-3,\,2q+1)}_{\text{cc}} + \Delta\phi^{(2q-3,\,2q+1)}_{\text{XUV}}$ and $\Delta\phi^{(2q-1,\,2q+3)}_{\text{Wig.}} + \Delta\phi^{(2q-1,\,2q+3)}_{\text{cc}} + \Delta\phi^{(2q-1,\,2q+3)}_{\text{XUV}}$. Again, the total phase can be well approximated by the average of these component phases, weighted by the amplitudes of the component signals. Figure 2i, shows that variation in the $4\omega_{\text{NIR}}$ phase, $\phi^{4\omega}_{\text{SB18}}$, in SB18 is similar to that of the $2\omega_{\text{NIR}}$ phase, $\phi^{2\omega}_{\text{SB16}}$, in SB16, but with an overall $\pi$ phase shift. This is because the $4\omega_{\text{NIR}}$ oscillation in SB18 includes the contributions of interference between (HH15 + 3)- and (HH19 - 1)-pathways, and between (HH17 + 1)- and (HH21 - 3)-pathways. Thus, because SB18 can be reached via the (resonant) HH15, its phase contains the signature of the Rydberg states[41]. The overall $\pi$ phase shift arises due to a sign difference in the SB18 $4\omega_{\text{NIR}}$ signal originating from interference between 4-photon and 2-photon paths, as opposed to interference between two, 2-photon paths in the SB16 $2\omega_{\text{NIR}}$ signal. Similar $\pi$ phase shifts have been noted in 'upper' and 'lower' sidebands within multi-sideband RABBITT experiments, for the same reasons[46].

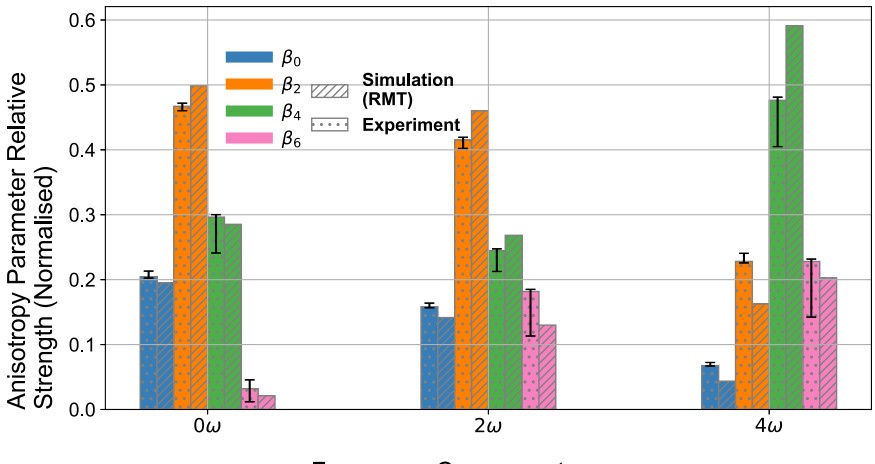

**Fig. 3 | Attosecond-resolved heterodyne analysis in helium.** The $0\omega_{NIR}$, $2\omega_{NIR}$ and $4\omega_{NIR}$ anisotropy parameter amplitudes which are extracted across SB16, normalised within each component signal. The anisotropy parameter amplitudes correspond to the fitted coefficients of the $0^{th}$ order (blue bars), $2^{nd}$ order (orange bars), $4^{th}$ order (green bars) and $6^{th}$ order (pink bars) Legendre Polynomials. The experimental relative amplitudes were obtained using a $1.1 \times 10^{12}$ W/cm² NIR (dotted bars), and the (RMT) simulated relative signal strengths were obtained using a $1.0 \times 10^{12}$ W/cm² NIR (lined bars). The experimental error bars account for both the $\pm 15°$ acceptance angle of the photoelectron detector, and the confidence in the fitting of the Legendre Polynomials to the PADs.

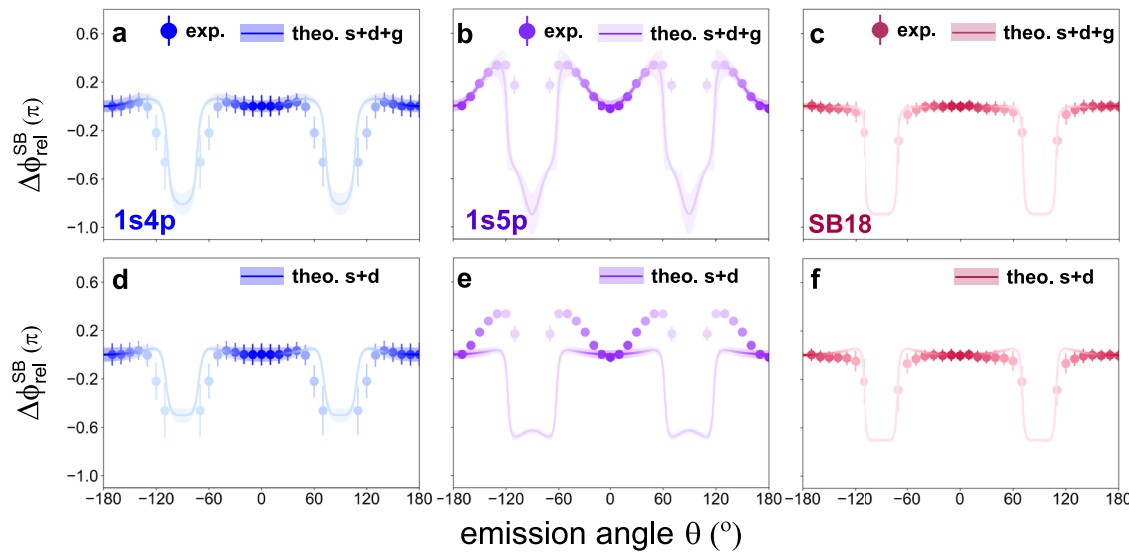

**Fig. 4 | Angle-resolved relative phase shifts. a–c** Angle-resolved phase shift distributions of the (**a**) SB16(4p), (**b**) SB16(5p) and (**c**) SB18 with a NIR intensity of $1.0 \times 10^{12}$ W/cm² (theory), $1.1 \times 10^{12}$ W/cm² (experiment). Solid circles are the experimental results, where the error bars indicate the standard deviation. Solid lines represent the theoretical simulations incorporating $s$, $d_0$ and $g_0$ waves, and the shaded areas indicate the error bars from the fitting procedure. **d–f** Same as (**a–c**) but only including the $s$ and $d_0$ wave interference in the RMT results.

Recent works have investigated high-order transitions with modified RABBITT techniques. One such scheme is multi-sideband RABBITT, where the driving NIR laser frequency is doubled before producing the XUV-APT such that two-photon transitions are effectively suppressed, revealing the four-photon ionization pathways[47,48], Other works have demonstrated the contribution of three-photon ionization to the main bands[32], and a general analysis of intensity resolved high-order contributions in PADs or RABBITT spectra[49–51]. However, none of the aforementioned works reveals the interaction between the contributing partial waves or their relative importance. In order to detect the contributions from high-order processes more accurately, we propose a heterodyne analysis based on the anisotropy parameters for the PADs from both experiment and RMT results, which reveals the role of interference between different partial

waves. The anisotropy parameters are the coefficients obtained by projecting the PADs onto a basis of Legendre Polynomials, which relate directly to the spherical harmonics comprising the measured signal[52–58]. Each partial wave has an associated spherical harmonic ($Y_l^m$), and the interference between a given pair of partial waves results in a PAD given by the product of their spherical harmonics. These products of spherical harmonics may also be expressed on a basis of spherical harmonics, with the property[59],

$$Y_{l_1}^0 Y_{l_2}^0 = \sum_{r=0}^{(l_1+l_2)} c_r Y_{l_1+l_2-r}^0, \tag{1}$$

where $\{c_r\}$ are expansion coefficients (where we note for even $(l_1, l_2)$ pairs, all odd-$r$ $c_r$ coefficients are zero), which can be mapped to their

corresponding anisotropy parameters $\{\beta_r\}$ (see Supplementary Fig. 2 in the SI). Partial waves beyond the $d$-wave appear through $\beta_n$ parameters with $n > 4$. The $\beta_6$ parameter will be characterized predominantly by interference between $d_0$ and $g_0$ waves. However, $\beta_8$ will contain contributions arising from $g_0 - g_0$ as well as $d_0 - i_0$ interference at the same order of magnitude. This mixing prevents use of the $\beta$ parameters to unambiguously characterize individual partial waves.

The relative strengths of anisotropy parameters from the $0\omega_{NIR}$, $2\omega_{NIR}$ and $4\omega_{NIR}$ oscillations in SB16 are shown in Fig. 3. For NIR intensities on the order of ~$10^{12}$W/cm$^2$, $\beta_6$ contributes to the $2\omega_{NIR}$ signal at the same level as $\beta_0$. Figure 3 demonstrates clearly that the angular distribution of the delay-averaged ($0\omega_{NIR}$) signal is well-described by interference involving only $s$ and $d_0$ waves as the $\beta_6$ component is negligible. However, the amplitude of $\beta_6$ in the $2\omega_{NIR}$ and $4\omega_{NIR}$ signals is approximately 10-20% of the total. The $\beta_8$ parameter is negligible in the $2\omega_{NIR}$ signal, which asserts that the overall magnitude of $g_0$ wave is small. Nevertheless, the $g_0$ wave interference with the other partial waves plays a key role in shaping the PADs.

To further identify the influence of the $g_0$ wave, we investigate the angular character of $2\omega_{NIR}$ oscillation phase shifts where the discrete SB16(4$p$) and SB16(5$p$) can be resolved in the experimental measurements, as shown in Fig. 4a/b for a NIR intensity of $10^{12}$W/cm$^2$. The result for the non-resonant SB18 is also displayed in Fig. 4c. The emission-angle-resolved relative phase shift of SB16 is defined as[5] $\Delta\phi_{rel}^{np}(\theta) = \phi_{np}^{2\omega}(\theta) - \phi_{np}^{2\omega}(\theta = 0^\circ)$. This cancels the chirp of the XUV field and normalizes the results. For $\Delta\phi_{rel}^{4p}(\theta)$ shown in Fig. 4a, the relative phase shift increases slightly to $0.05\pi$ as the emission angle deviates from $0^\circ$ and then rapidly drops towards $\theta = 90^\circ$. However, the angular distribution of $\Delta\phi_{rel}^{5p}(\theta)$ in Fig. 4b displays a more significant angular variation with $\Delta\phi_{rel}^{5p}(\theta)$ increasing to $0.36\pi$ at $\theta = 60^\circ$. If we use only the $s$ and $d_0$ waves from the high-intensity calculation, the $\Delta\phi_{rel}^{5p}(\theta)$ is not well described (Fig. 4e). The inclusion of the $g_0$-wave is thus crucial to recover the correct angular behavior. Meanwhile, the different angular structures of $\Delta\phi_{rel}^{4p}(\theta)$ and $\Delta\phi_{rel}^{5p}(\theta)$ demonstrate that the transition amplitude and phase of the bound-continuum transition are dependent on the specific Rydberg states involved. For the non-resonant SB18, the interference among $s$, $d_0$ and $g_0$ waves displays a slight modulation in the $2\omega_{NIR}$ phase shift distributions but also gives a better agreement with the experimental results. This suggests that the NIR-assisted ionization from Rydberg states acts as a more sensitive probe of the intensity variation. The emission-angle-resolved phase shifts with a higher NIR intensity of 3.2 TW/cm$^2$ are shown in Supplementary Fig. 3 in the SI.

## Discussion

Importantly, the PAD encodes the interference between the individual partial waves, information that is lost in the integrated spectrum. The RMT simulation suggests that the $g_0$ wave contributes only on the order of 1% of the total sideband yield. This supports our earlier assertion based on the analysis of the anisotropy parameters, and we can thus surmise that it is the interference between the $g_0$-wave and the $s$- and $d_0$-waves which gives rise to the characteristic angular distributions, especially in $\Delta\phi_{rel}^{5p}(\theta)$. We also note that, given the sensitivity of the angular distribution to the small $g_0$-wave contribution, the spectral positions of not only the 1s4$p$ and 1s5$p$ resonances, but also of the 1s$nf$ Rydberg states, are important. The absorption of one HH13-photon together with two NIR photons has the same resultant photon energy as a HH15-photon, but the indirect pathway may populate the 1s$nf$ states.

In summary, we have explored the RABBITT technique with a TW-level NIR pulse to reveal the role of high-order processes (RABBITT-HOP). Fourier transforming the RABBITT-spectrum from a few-photon ionization of He reveals a clear oscillation at four times the energy of the NIR field, showing unambiguously the presence of four-photon ionisation pathways. A comprehensive analysis using anisotropy parameters extracted from the PADs is performed, where those parameters encoding interference amongst the $s$, $d_0$, $g_0$ and $i_0$ partial waves are shown to contribute at the ~10−20% level. By leveraging the interference, rather than the direct contributions, this technique is a heterodyne measurement of the otherwise undetectable high-order partial waves. We further demonstrate the contribution of the $g_0$ wave to the $2\omega_{NIR}$ oscillation by considering the angular distributions. Simulations neglecting the role of the $g_0$ partial wave– which can arise only via the absorption of four photons– fail to reproduce the unusual, angle-resolved phase shift distribution. While simulations reveal that the $g_0$-wave comprises only ~1% of the total sideband yield, its interference with the lower-order partial waves is crucial for reproducing the experimentally measured phase.

Our findings suggest that even for comparatively low NIR intensities ($10^{12}$ W/cm$^2$) few-photon processes are non-negligible in attosecond photoionization of atoms, especially for RABBITT experiments involving resonant channels, and will affect the experimental observable by populating partial waves with higher angular momentum than two-photon transitions. This RABBITT-HOP analysis extends attosecond-resolved studies of photoelectron dynamics, helping to identify the contributing transition pathways as well as providing a means to quantify the contribution from high-order partial waves. Thus, this heterodyne method paves the way to understand the fundamental transition from the perturbative ionization regime to strong-field ionization[60−62] and provides a general, partial-wave-resolved method to investigate multi-photon ionization time delays.

## Methods

### Attosecond coincidence interferometer

The angle-resolved, XUV-NIR delay-dependent photoelectron spectra are measured in a $4\pi$ solid angle via an attosecond coincidence interferometer[4,5]. A multi-pass amplified Ti:Sapphire laser system is used to produce a near-infrared ~775 nm femtosecond laser pulse with a pulse duration of 28 fs at a repetition rate of 10 kHz. This NIR pulse is split in two to construct a nonlinear Mach-Zehnder interferometer scheme. One component of the split NIR pulse is focused into a capillary filled with krypton gas to generate the XUV-APT with a high harmonic comb covering 13th order (HH13, 20.8 eV) to 19th order (HH19, 30.4 eV). The resulting XUV-APT passes through a coaxial aluminum foil installed on a quartz ring with a thickness of 100 nm to block the driving NIR pulse and is further focused by a toroidal mirror. The second component of the NIR pulse serves as the probe arm, and travels through a half wave-plate to ensure the polarization axis is parallel to the XUV-APT. The two arms of the nonlinear interferometer propagate co-linearly after recombination by a holey sliver mirror, and are focused into a supersonic helium gas jet in the reaction chamber of a COLTRIMS set-up[29,30]. The relative phase-locking between the XUV-APT and NIR field is realized through an interferometer scheme where a 532 nm continuous wave (CW) beam is split to co-propagate with the two fields. After the recombination through the holey mirror, the interference of the two beams form radial fringes[61,63]. A slight fluctuation of path length leads to the position shift of the fringes. During the actively stabilized phase scanning, the position of selected radial fringes is locked by a proportional-integral-derivative (PID) algorithm using a precise piezo stage with an average time jitter of $\pm 28$ as. The coincidence measurement of ion fragments and photoelectrons is realized by a COLTRIMS set-up; the ionization produces ionic fragments and photoelectrons, which are guided by a homogeneous electric and magnetic field towards ion and electron detectors (respectively), installed in opposing directions. Both detectors contain two micro-channel plates (MCPs) and a delay-line anode to precisely record the position and time-of-flight of the particles, and are further used to trace back to the three–imensional momenta at the moment of ionization.

## Theoretical RMT model

The theoretical results were obtained using the R-Matrix with Time-dependence (RMT)[26–28] approach. The RMT method simulates the interaction between a single atom, in this case, helium, with an external laser field while accounting for electron correlation effects. RMT employs a division of the interaction space into an 'inner' region spanning a radius of 15 a.u. from the nucleus (large enough to contain the residual ion) and an 'outer' region (where only a single active electron may reside) extending to 6,226 a.u. Within the inner region, full account is taken of electron exchange, while in the outer region the electron interacts with the residual ion only via long-range potentials. Interaction with the laser field is modeled under the dipole approximation, and the system is evolved in time by solving the time–ependent Schrödinger equation (TDSE) in each region and matching the wavefunction at the boundary. This division permits a highly accurate solution of the TDSE for a greatly reduced computational cost compared to models which account for multi-electron effects throughout the full interaction space. We employ the '1T' atomic description for helium (see refs. 5,64) within our simulations. We model the RABBITT process with a 775.5 nm NIR field with a peak intensity of 1.0 TW/cm$^2$ comprising 14 cycles, of which the first and last 3 are sin$^2$ 'ramp on/off' cycles, and a XUV-APT field of peak intensity 0.02 TW/cm$^2$ and comprising 8 bursts, of which the first and last two are 'ramp bursts', with frequency components of odd harmonics of the NIR field centered around HH15. We simulate 32 evenly spaced delays between the XUV-APT and NIR fields, spanning one NIR period, each for a total simulation time of 96.76 fs. We then decouple the (position-space) photoelectron wavefunctions from the residual ion and project them into momentum space via Coulomb-distorted waves to account for the effect of the residual ion on low-energy components, and obtain the momentum space angular distribution through convolution with the complex spherical harmonics associated with each component partial-wave[65].

## Data availability

The data generated in this study have been deposited in the Zenodo database under accession code:https://doi.org/10.5281/zenodo.13331808.

## Code availability

The RMT theoretical simulation codes are shared in gitlab (https://gitlab.com/Uk-amor/RMT/rmt), and the other codes that support the findings of this study are available from the corresponding author upon request.

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

## Acknowledgements

This work was supported by the National Natural Science Foundation of China (Grant Nos. 12241407, 12261160363, 12122404, 11974114, 12227807), the Science and Technology Commission of Shanghai Municipality (Grant No. 23560760100, 228002246, 19560745900, 23JC1402000), the Shanghai Pilot Program for Basic Research (Grant No. TQ20240204), and the Fundamental Research Funds for the Central Universities. W.J. acknowledges the support from the National Post-doctoral Program for Innovative Talents (Grant No. BX20240333). L.R., A.B. and H.v.d.H. acknowledge the work from Gregory S.J. Armstrong, Daniel D. A. Clarke and Jakub Benda in developing new capabilities within the RMT code which were used in this work, and the funding from the UK Engineering and Physical Sciences Research Council (EPSRC) under grants EP/T019530/1, EP/V05208X/1 and EP/R029342/1. This work relied on the ARCHER2 UK Supercomputer Service, for which access was obtained via the UK-AMOR consortium funded by EPSRC. The RMT code is part of the UK-AMOR suite. This work benefited from computational support by CoSeC, the Computational Science Centre for Research Communities, through CCPQ.

## Author contributions

W.J., L.H., X.G., and J.W. performed the experiments, L.R., A.B., and H.v.d.H., performed the theoretical calculations. W.J., L.R., L.H., Y.X. and Z.Z. analysed the data. J.J., K.U., and H.J.W. participated in the fruitful discussions. W.J., L.R., L.H., A.B., K.U., H.v.d.H., X.G. and J.W. wrote the paper with input from all co-authors.

## Competing interests

The authors declare no competing interests.
