## [Transparent Peer Review file · Nature Communications]

Heterodyne analysis of high-order partial waves in attosecond photoionization of helium

Corresponding Author: Professor Jian Wu

Version 0:

Reviewer comments:

Reviewer #1

(Remarks to the Author)

Helium atoms are photoionized by the harmonic orders of an attosecond pulse train, in the presence of a synchronized infrared beam. The kinetic energy and angular distribution of the photoelectrons are recorded by a COLTRIMS device. As the delay between the XUV and NIR fields is varied, the intensity of the photoelectron angular distributions changes. A model based on the time-dependent R-matrix method shows good agreement with the experiment.

The manuscript is well written and clear. I support the acceptance of the manuscript.

Some specific comments:

- (1) In Figure 2e, the main photoelectron peaks corresponding to direct 1-photon ionization do not appear. Why does the RMT model not produce these?
- (2) In Figure 2a, I cannot see the two frequencies of SB16 that are evident in Figure 2e. Yet I think in Figure 2b, the 4-omega peak is at a lower frequency than the 2-omega peak. Should I be able to see the two peaks in 2a? What is the significance of the 2-omega and 4-omega peaks being associated with the 4p and 5p channels?
- (3) Mayer et al. (J Phys B 53, 164003 (2020)) report similar experiments using a VMI instead of a COLTRIMS, but use a higher IR intensity than in the present manuscript. They show similar multiphoton pathways from the high-lying states of helium to the continuum. Please comment on how the RMT model compares with the TDSE model of Mayer.
- (4) An in-situ technique can also be used to measure the atto-chirp of a pulse. For a higher perturbing intensity, the sidebands and main bands no longer oscillate exactly out of phase. In the present experiment at higher NIR intensity, was there any evidence that the sidebands and main bands are not out of phase? For example, Figure 4b shows a phase shift at 90 degrees that is greater than π , and less than π in Figure 4f.
- (5) On page 9, "The PAD encodes the interference between the individual partial waves". The phases shown in Figure 4 are the XUV-NIR delay phases, not the partial wave phases. Does the XUV-NIR delay scan allow one to determine the relative photoionization phases of the various partial waves, based on the spherical harmonic decomposition?
- (6) Was the fitting to spherical harmonics done with the full 3D photoelectron angular distribution, or was it flattened into a 2D ring and fit with Legendre polynomials? Does the third dimension contain any information that is not found with VMI detection?
- (7) The different pathways to SB16 illustrated in Figure 1 involve ionization with different harmonic orders. For example, SB16 can be reached from the 4p and 5p bound states, or from HH17, or from HH19. The continuum-continuum coupling will be different for each of these channels. Is this significant?
- (8) Figure 1 refers to a "grey shadow". It looks like a pink shadow to me.
- (9) The wavelength of the main laser (775 nm) should be included on page 4.

(10) Figure 2d, the y-axis is $\lg(A)$, and is called "logarithmic" in the caption. Please clarify if this is the natural logarithm or base-10.

Reviewer #2

(Remarks to the Author)

The manuscript "Heterodyne analysis of high-order partial waves in attosecond photoionization of Helium" by W. Jiang et al. demonstrates that angle-resolved phase shifts obtained in a RABBITT experiment on He are a sensitive probe for the contribution of high-order partial waves because of multi-photon transitions. The phases (and photoionization time delays) extracted from RABBITT experiments could be falsified if the strict two-photon transition scheme of RABBITT is not kept. The proposed heterodyne analysis taking advantage of angularly-resolved photoelectron detection shows a clever approach beyond a simple FFT analysis to detect oscillations with a frequency of 4ω how to verify if few-photon transitions play a role. The manuscript is well written, the presented experimental and computed data is of high quality and the analysis methodology is sufficiently well explained to be able to reproduce this work. The conclusions drawn from the experimental and theoretical results are presented in a convincing way. My recommendation is to accept this manuscript for publication in Nature Communications.

However, I have some remarks/questions on the manuscript:

1) Figure 1b is very dense in information. Especially the meaning of the blue and green shadows for the 2ω and 4ω oscillations are not really clear for me. The origin of the two types of oscillations (i.e. which bands are participating) is well explained in the manuscript, but either a further explanation, e.g., in the figure caption, is necessary or, since I don't have the feeling that I am missing information, these shadows are unnecessary for the comprehension of the experiment and might become confusing (as for me).

2) On page 6, second paragraph, it is stated that the contributions of the interferences a) and b) are of similar magnitude. Is this the case because the intensity of HH($2q-1$), HH($2q+3$), HH($2q-3$), HH($2q+1$) is either flat or symmetric? In Fig. 2a it seems that H17 and H19 have quite different intensities. How does is the intensity of H15 and H21 (if we look at SB18)?

3) The involvement of g-waves, which indicate that >2 photons are involved, is only visible in the angle-resolved phase shift (Fig. 4) if a Rydberg state is involved. Does this mean that in the absence of resonances in the photoionization process, the extracted phases/delays are not compromised if one has slightly passed the perturbative regime yet?

4) In the supplementary material the oscillation of the anisotropy parameters as a function of the XUV-NIR delay is shown (among others) for computations at different NIR intensities. It would have been interesting to see the transition to the strong-field regime for experimental data in which the intensity is varied and the observed effect in the angle-resolved phase for SB16(5p) can be "switched on" by a certain threshold intensity.

Reviewer #3

(Remarks to the Author)

The manuscript "Heterodyne analysis of high-order partial waves in attosecond photoionization of Helium" presents measurements and simulations of an angle-resolved RABBITT setup in helium, showing that contributions of higher-order transitions (i.e., beyond two photons) can be observed in the higher-order anisotropy parameters of the photoelectron angular distribution. Particular emphasis is placed on the fact that the anisotropy parameters arise due to the interference of different partial waves, such that they can be more sensitive to the (small) amplitudes of higher partial waves than their associated probabilities would be. While the results and analysis seem correct, these physical concepts are well-known and understood, and the manuscript does not show whether this approach gives any new information on the underlying physics of the process. While the work is thus a proof-of-concept demonstration of the idea, it does not go beyond this at all, and the possible impact or potential of the method remains unclear. I thus believe that a more specialized journal such as Physical Review A would be more appropriate for this work.

For resubmission to any journal, the following more specific points should be addressed:

-) Overall, the manuscript is not particularly well written. There are seemingly contradictory statements, e.g., in the introduction "... at higher intensities, higher-order partial waves are unavoidable through multi-photon transitions. In this case, the contribution of higher-order partial waves should be dominated by those arising from one or two-photon transitions, making their detection impossible." - so are they unavoidable or impossible to detect? The manuscript should be carefully proofread and revised for clarity.

-) The schematic in Fig. 1 is quite unclear: The thick black line is labeled both as $1s^2$ and as I_p . The "grey shadow" is almost invisible. The meaning of the blue and green regions and the diffuse lines and short markers inside them is not explained.

-) The theory results show a clear double peak for SB16, while nothing of the like is visible in the experimental data. This discrepancy should be discussed.

-) It is claimed that when there are several contributions at the same oscillation frequency to a given sideband, "the total

phase is the average of these component phases weighted by the amplitudes of the component signals". This is not correct. The sum of two sines with different phases and weights does not give a sine with the weighted average of those phases.

-) The explanation that the product of spherical harmonics can be expressed in the basis of spherical harmonics is well-known undergraduate-level material and its discussion could be significantly shortened. It is similarly well-known that the angular distributions depend on the interference between partial waves.

Version 1:

Reviewer comments:

Reviewer #1

(Remarks to the Author)

The authors have satisfactorily answered all of the referees' questions. I recommend that the present manuscript be accepted for publication.

Reviewer #2

(Remarks to the Author)

The authors undertook some effort to respond to all points raised by the referees. My comments were addressed diligently and the revision of the text and the figures leads to more clarity for the reader. Additional experimental results were included in the Supporting Information, which will also help the interested reader to dive deeper into the interesting results presented in this paper.

The study shows interesting new aspects on RABBITT experiments with stronger NIR fields and also lets the community reconsider older published experiments where the influence of higher order partial waves was neglected. Therefore, this paper can be considered impactful enough to justify a publication in Nature Communications.

Reviewer #3

(Remarks to the Author)

The authors have carefully answered all points raised by the referees, and have significantly improved the manuscript. While I am still not fully convinced that this work rises to the level of Nature Communications, I think it is certainly a nice contribution and I am happy to join the other two referees in recommending publication.

We thank the referees for reviewing our manuscript, and for their thoughtful comments and suggestions. We have revised the manuscript point by point according to their recommendations. Below, we reproduce the referee comments in black, provide our answers in blue, and present the changes made to the manuscript in green.

 Report of Referee #1 -- NCOMMS-24-32814

Helium atoms are photoionized by the harmonic orders of an attosecond pulse train, in the presence of a synchronized infrared beam. The kinetic energy and angular distribution of the photoelectrons are recorded by a COLTRIMS device. As the delay between the XUV and NIR fields is varied, the intensity of the photoelectron angular distributions changes. A model based on the time-dependent R-matrix method shows good agreement with the experiment.

The manuscript is well written and clear. I support the acceptance of the manuscript.

We thank the referee for their summary and kind recognition.

Some specific comments:

(1) In Figure 2e, the main photoelectron peaks corresponding to direct 1-photon ionization do not appear. Why does the RMT model not produce these?

We are thankful for the referee's valuable comments.

The original Figure 2e shows the signal calculated using only the s, d and g partial waves from RMT, thus omitting the main peaks. We have revised this figure on page 6 by including the theoretical main-peak signal to avoid confusion.

Figure R1: Attosecond photoelectron kinetic energy spectrum. **a** Experimentally measured photoelectron kinetic energy spectrum; shadows represent the results of multi-peak Gaussian

fitting. **b** Attosecond-resolved photoelectron spectrum with a NIR intensity of $1.1 \times 10^{12} \text{W/cm}^2$. **c**, **d** FFT amplitude spectrum and phase shift distributions. In **d**, blue and the green dots represent phase shifts of $2\omega_{\text{NIR}}$ and $4\omega_{\text{NIR}}$ oscillations, respectively. **e** The gated amplitude distributions at $2\omega_{\text{NIR}}$ (blue area) and $4\omega_{\text{NIR}}$ (green area) on a base-10 logarithmic scale. **f-j** As **a-e**, but for RMT simulations with a NIR intensity of $1.0 \times 10^{12} \text{W/cm}^2$. Further analysis of partial wave contributions is given in the Supplementary Information. (This figure is the same as Fig. 2 in the revised manuscript.)

(2) In Figure 2a, I cannot see the two frequencies of SB16 that are evident in Figure 2e. Yet I think in Figure 2b, the 4-omega peak is at a lower frequency than the 2-omega peak. Should I be able to see the two peaks in 2a? What is the significance of the 2-omega and 4-omega peaks being associated with the 4p and 5p channels?

As noted, our RMT simulation allows us to clearly resolve the 4p and 5p contributions. In experimental measurements, however, the distinction between 4p and 5p channels is limited by the kinetic energy resolution. Despite this, the delay-averaged photoelectron kinetic energy spectrum (from the experiment) and the 2ω phase shifts across SB16 show a clear signature of the two Rydberg states. To clearly display this result, we have modified Fig. 2 by adding the photoelectron kinetic energy spectrum (Fig. R1a) and a “zoom-in” (Fig. R1d) of the experimental 2ω phase shifts across SB16, as shown in Fig. R1.

The following sentence has been added in the second paragraph on page 5:

“This double peak structure is visible in the theoretical results in Fig. 2g but is obscured by the relatively lower kinetic energy resolution in the experimental spectra in Fig. 2b. However, the interference also modifies the phase of the resulting $2\omega_{\text{NIR}}$ oscillation, revealing the signature of photoionization from the double Rydberg states which can be seen both in the inset in Fig. 2d and in Fig. 2i [Ref. 37, Ref. 40, Ref. 41].”

We identify the 4p- and 5p-Rydberg states imprints in SB16 from the peaks in the 0ω (i.e. delay-averaged) spectrum, (shown in Fig R1 **a** and **f**), which align closely with the peaks in the 2ω spectrum. However, the 4ω oscillations in SB16 resulted from the interference between (HH13 + 3NIR) and (HH17 - 1NIR), and between (HH15 + 1NIR) and (HH19 - 3NIR). Thus, it includes the coupled contributions not only from the 4p and 5p but also the nf-Rydberg states. As the referee notes, the 4ω spectrum appears slightly shifted in kinetic energy from the experimental measurements. We have not been able to confidently identify the origin of this shift, but we believe it warrants further study in a future investigation.

While the RMT results resolve the 4p and 5p contributions even in the photoelectron yield, the main evidence for their distinct contributions in the experimental data comes from the different angular distributions of the 2ω phase shifts. These results thus provide direct evidence that the transition amplitudes and phases of bound-continuum transitions depend on the intermediate Rydberg states.

The following sentence was added in the first paragraph on page 10:

“Meanwhile, the different angular structure of $\Delta\phi_{\text{rel}}^{4p}$ and $\Delta\phi_{\text{rel}}^{5p}$ demonstrates that the transition amplitude and phase of the bound-continuum transition is dependent on the specific Rydberg states.”

(3) Mayer et al. (J Phys B 53, 164003 (2020)) report similar experiments using a VMI instead of a COLTRIMS, but use a higher IR intensity than in the present manuscript. They show similar multiphoton pathways from the high-lying states of helium to the continuum. Please comment on how the RMT model compares with the TDSE model of Mayer.

The calculations are the same at the fundamental level, and we expect that the insights in this particular case would be the same. RMT is more flexible: Muller’s TDSE code is specific to helium and uses the single active electron approximation on a 2D grid, whereas RMT can be applied to general multielectron atoms and molecules. The RMT approach uses a basis-set approach in the inner region. This basis-set approach makes it easy to shift the ground-state energy to the experimental position.

A reference to the Mayer et al. paper has been added into the revised manuscript as [Ref. 51].

(4) An in-situ technique can also be used to measure the atto-chirp of a pulse. For a higher perturbing intensity, the sidebands and main bands no longer oscillate exactly out of phase. In the present experiment at higher NIR intensity, was there any evidence that the sidebands and main bands are not out of phase? For example, Figure 4b shows a phase shift at 90 degrees that is greater than π , and less than π in Figure 4f.

Figure R2: Experimental photoelectron kinetic energy spectra and 2ω phase shifts. **a** Experimental results for SB16. **b** Experimental results for MB17. **c** RMT results for SB16. **d** RMT results for MB17.

We thank the referee for their constructive ideas. If we understand the referee's idea correctly, they point out that the phase of the 2ω signal in SB16 encodes the phase differences of ionization paths via the HH15 and HH17 harmonics, that these differences are similarly encoded in HH17, but (due to the interference now being between 1- & 3-photon paths rather than 2- & 2-photon paths in the sideband) an additional π phase shift is induced.

While this is the case, the 2ω signal in HH17 also includes contribution from HH17-HH19 interference and thus we would expect the HH17 2ω phase to deviate a little from the (π shifted) SB16 2ω phase. This is discussed in more detail in Drescher 2022 (DOI: 10.1103/PhysRevA.105.L011101) and Bharti 2021 (DOI: 10.1103/PhysRevA.103.022834).

A derivation has been added to the page 8 of the Supplementary Information to demonstrate how a main-band's total 2ω phase may be constructed from its interferences with the higher and lower main bands, and we note that the NIR intensity dependence of this behavior will be explored in much greater detail in a planned follow-up manuscript.

Measuring the kinetic energy resolved HH17 2ω phase and comparing it with the SB16 2ω phase (shown in Figure R2 above; panels a), b) showing experimental results and c), d) showing RMT simulation results), we observe strong qualitative similarity in the shape of the encoded phases, with a -0.9π shift, close to the π phase shift the referee suggests.

Figure 4 in the revised manuscript displays the $\Delta\phi_{\text{rel}}^{\text{SB}}(\theta)$ with respect to $\Delta\phi_{\text{rel}}^{\text{SB}}(\theta = 0^\circ)$ for SB16(4p), SB16(5p) and SB18. The different angular structure originates from the partial wave interference among s, d, g and even i waves with different partial wave proportions and relative phase shifts.

(5) On page 9, "The PAD encodes the interference between the individual partial waves". The phases shown in Figure 4 are the XUV-NIR delay phases, not the partial wave phases. Does the XUV-NIR delay scan allow one to determine the relative photoionization phases of the various partial waves, based on the spherical harmonic decomposition?

We thank the referee for this thoughtful idea. In principle, one can reconstruct the partial wave amplitudes and phases by fitting the PADs and the emission angle resolved phase shifts, provided you have sufficiently resolution in the PADs to eliminate unknown parameters in the spherical harmonic expansions. In previous works, we accomplished this using different relative polarization angles between the NIR and XUV-APT fields. We did attempt the reconstruction in an earlier version of the current work, but it was found that the fitting was incredibly sensitive to the small amplitude terms in the expansion of the spherical harmonics, and the phase values we recovered were thus unreliable. The analysis of the β parameters, by contrast, is very robust but comes at the cost of 'mixing' the interference terms.

The following sentence has been added into page 5 in the revised Supplementary Information:

"In the general RABBITT measurements within the perturbative regime, the partial-wave resolved two-photon phase shifts and proportions can be reconstructed by fitting the PADs and the emission-angle resolved phase shift distributions based on the spherical harmonics [Ref. 7 and Ref. 8]. However, as the few-photon transitions are involved, this kind of fitting is unstable due to the

tiny amount of the high-order partial waves, g_0 - and i_0 -waves here, thus adding the uncertainty of the reconstructed phase shifts. In this case, the heterodyne analysis of anisotropy parameters serves as a sensitive tool to detect the participation of high-order partial waves.”

(6) Was the fitting to spherical harmonics done with the full 3D photoelectron angular distribution, or was it flattened into a 2D ring and fit with Legendre polynomials? Does the third dimension contain any information that is not found with VMI detection?

We use the flattened, 2D spectra for the fitting of measured experimental data.

In simulation, we evaluate a 2D ‘slice’ of the full 3D angular distribution at $\pi/2$ radians on the azimuthal axis. In the experiment, the full 3D angular distribution was recorded, and a similar 2D slice was used in the fitting procedure. We define an angle $\theta_x = \arcsin(p_{ex}/p_e)$ and choose the photoelectrons which satisfy the condition with $|\theta_x| < 30^\circ$.

The use of the 2D slices allowed for direct projection onto Legendre polynomials rather than full spherical harmonics, simplifying the analysis procedure. As linear parallel polarized laser fields were used in this investigation, there should be no photons with angular momenta of ± 1 available to populate non-zero magnetic sublevels. In this case, the 3D angular distributions should be azimuthally symmetric and we should lose no information by considering only the 2D slices. In recently published variations of RABBITT, such as W. Jiang et al. (DOI:10.1038/s41467-022-32753-8) in 2022 and M. Han et al. (DOI:10.1038/s41567-022-01832-4) in 2024, this azimuthal symmetry is lost and fitting of the full 3D angular distribution to spherical harmonics would be required for a similar analysis. However, this would certainly be achievable as an extension of our proposed heterodyne detection method, and we hope this paper inspires such an approach.

(7) The different pathways to SB16 illustrated in Figure 1 involve ionization with different harmonic orders. For example, SB16 can be reached from the 4p and 5p bound states, or from HH17, or from HH19. The continuum-continuum coupling will be different for each of these channels. Is this significant?

We thank the referee for their thoughtful idea. The continuum-continuum coupling is different in each of these channels. An efficient asymptotic approximation far beyond the near-threshold ionization was proposed by J. M. Dahlström in 2013 (DOI: 10.1016/j.chemphys.2012.01.017) to simplify the understanding of these couplings, where the absolute CC phase for absorption and emission pathways is identical. Further experimental work by J. Fuchs et al. in 2020 (DOI: 10.1364/OPTICA.378639) demonstrated that the CC phase is also dependent on the angular momenta. However, these results were all focused on photoelectrons with a kinetic energy above 4 eV. In our manuscript, these channel-resolved continuum-continuum couplings in near-threshold ionization and the bound-transitions from Rydberg states are intrinsically accounted for both the RMT *ab initio* simulations and experimental measurements. Thus, we believe that the good agreement between experimental results and theoretical simulations proves that our theoretical model may be a useful tool for further investigation into the channel-resolved CC couplings.

(8) Figure 1 refers to a “grey shadow”. It looks like a pink shadow to me.

We are grateful for the suggestion.

Figure 1 has been revised as the following Fig. R3, the “shadow” is removed to avoid confusion.

Figure R3: Schematic diagram of the attosecond coincidence interferometer. **a** Experimental configuration. **b** Transition map of photoionization in helium atoms. In the two-photon regime, the interference between HH(2q-1) and HH(2q+1) pathways via absorption and emission of one NIR photon leads to a $2\omega_{\text{NIR}}$ oscillation in the yields of SB(2q). Extending into the four-photon regime, the interference between (HH(2q-3)+3NIR) and (HH(2q-1)+NIR) pathways, (HH(2q-1)+NIR) and (HH(2q+1)-NIR) pathways, (HH(2q+1)-NIR) and (HH(2q+3)-3NIR) pathways also contribute to the $2\omega_{\text{NIR}}$ oscillations, where the plus (minus) symbol represents the absorption (emission) of NIR photons. The additional $4\omega_{\text{NIR}}$ oscillations arise from the four-photon-transition-induced interference between (HH(2q-3)+3NIR) and (HH(2q+1)-NIR), or (HH(2q-1)+NIR) and (HH(2q+3)-3NIR) pathways. (the same as Fig. 1 on page 3 in the revised maintext.)

(9) The wavelength of the main laser (775 nm) should be included on page 4.

The wavelength has been added on page 4 in the revised main text.

“As illustrated in Fig. 1a, the experimental measurements were performed using an attosecond coincidence interferometer— based on a near-infrared 775 nm femtosecond laser pulse— with a high momentum resolution for photoelectron kinetic energy below 7 eV.”

(10) Figure 2d, the y-axis is $\lg(A)$, and is called “logarithmic” in the caption. Please clarify if this is the natural logarithm or base-10.

We thank the referee for their kind advice, and this has been clarified on page 6.

“e The gated amplitude distributions at $2\omega_{\text{NIR}}$ (blue area) and $4\omega_{\text{NIR}}$ (green area) on a base-10 logarithmic scale.”

The manuscript "Heterodyne analysis of high-order partial waves in attosecond photoionization of Helium" by W. Jiang et al. demonstrates that angle-resolved phase shifts obtained in a RABBITT experiment on He are a sensitive probe for the contribution of high-order partial waves because of multi-photon transitions. The phases (and photoionization time delays) extracted from RABBITT experiments could be falsified if the strict two-photon transition scheme of RABBITT is not kept. The proposed heterodyne analysis taking advantage of angularly-resolved photoelectron detection shows a clever approach beyond a simple FFT analysis to detect oscillations with a frequency of 4ω how to verify if few-photon transitions play a role.

The manuscript is well written, the presented experimental and computed data is of high quality and the analysis methodology is sufficiently well explained to be able to reproduce this work. The conclusions drawn from the experimental and theoretical results are presented in a convincing way. My recommendation is to accept this manuscript for publication in Nature Communications.

We thank the referee for their kind comment and recognition.

However, I have some remarks/questions on the manuscript:

1) Figure 1b is very dense in information. Especially the meaning of the blue and green shadows for the 2ω and 4ω oscillations are not really clear for me. The origin of the two types of oscillations (i.e. which bands are participating) is well explained in the manuscript, but either a further explanation, e.g., in the figure caption, is necessary /or, since I don't have the feeling that I am missing information, these shadows are unnecessary for the comprehension of the experiment and might become confusing (as for me).

We thank the referee for the kind suggestions. Fig. 1 on page 3 has been modified to remove all shadows, and detailed descriptions of the 2ω and 4ω oscillations has been added to the caption. The figure and caption are reproduced as Fig R1 above for convenience.

2) On page 6, second paragraph, it is stated that the contributions of the interferences a) and b) are of similar magnitude. Is this the case because the intensity of HH($2q-1$), HH($2q+3$), HH($2q-3$), HH($2q+1$) is either flat or symmetric? In Fig. 2a it seems that H17 and H19 have quite different intensities. How does is the intensity of H15 and H21 (if we look at SB18)?

We thank the referee for their careful review. The experimental and theoretical photon energy spectra are shown in Fig. R4. The experimental spectrum is reconstructed by shifting the photoelectron kinetic energy spectrum of neon atoms with its ionization potential, and the relative intensity of harmonic combs is further calibrated using the one-photon-ionized cross-section. In our simulations, the XUV spectrum is indeed (by construction) symmetric about HH15 (i.e. HH($2q-1$) in this section of the manuscript), as shown in Fig. R4b below. In our simulations, we use an attosecond pulse train comprising 9 full-strength attosecond bursts, such that the HH13(HH17) strength is 64% of HH15 and HH11(HH19) is 16%.

While this results in HH15-HH19 interference and HH13-HH17 interference contributing in a 1:2 ratio, our claim that the two sources contribute with similar magnitudes is within the context of the overall 4ω signal contributing around two orders of magnitudes less than the main 2ω signal. It has not been possible to accurately resolve the strength of HH13 in the experiment, preventing a similar comparison of the HH15-HH19 and HH13-HH17 contributions, but we expect we would obtain a similar ratio. The use of a longer attosecond pulse train would cause the relative contributions of the two sources to converge further. Within SB18, the main two sources of 4ω signals are HH17-HH21 interference and HH15-HH19 interference, which contribute to a 1:19 ratio.

To clarify this confusion, we modified the sentence in the last paragraph on page 5 as follows: “Most notably, the $4\omega_{\text{NIR}}$ yield contains contributions of from a) interference between ionization paths $(\text{HH}(2q-1) + 1)$ and $(\text{HH}(2q+3) - 3)$, and b) interference between $(\text{HH}(2q-3) + 3)$ and $(\text{HH}(2q+1) - 1)$ where the plus (minus) symbol represents absorption (emission) of NIR photons.”

Figure R4: XUV spectra. **a** Experimental spectrum calibrated from the photoelectron energy (PE) spectrum of neon atoms. The green dots represent the original one-photon-ionized PE spectrum of neon shifted by its ionization potential. The blue line shows the experimental XUV spectrum calibrated using the one-photon ionization cross section of neon. **b** Simulated XUV energy spectrum showing harmonics HH9 (14.43eV) to HH21 (33.65eV).

3) The involvement of g-waves, which indicate that >2 photons are involved, is only visible in the angle-resolved phase shift (Fig. 4) if a Rydberg state is involved. Does this mean that in the absence of resonances in the photoionization process, the extracted phases/delays are not compromised if one has slightly passed the perturbative regime yet?

From our results, SB18 (which may be reached from the resonant states only via $(\text{HH15} + 3)$ -photon paths) is indeed less sensitive to the NIR intensity than SB16 (which may be reached via single-photon transitions from the resonant states), and thus a first reading of these results would suggest that the resonance somehow promotes the higher order processes. Intuitively, enhancing specific pathways via a resonance should increase the probability that those pathways would contribute. To answer this and the next question more clearly, we performed additional experimental measurements with a higher NIR intensity, $I_{\text{NIR}} = 3.2 \text{ TW/cm}^2$. In this case, the angular structure of the 2ω phase shift in SB18 is modulated, as shown in Fig. R5. Thus, in the absence of a more general survey of different systems, it would be premature to say that it is

necessary.

Figure R5: The emission-angle-resolved $2\omega_{\text{NIR}}$ phase shifts with a NIR field intensity of 3.2 TW/cm^2 as measured in the experiment: (a) SB16(5p), (b) SB18. (the same as Supplementary Fig. 3 on page 8 in the supplementary information.)

4) In the supplementary material the oscillation of the anisotropy parameters as a function of the XUV-NIR delay is shown (among others) for computations at different NIR intensities. It would have been interesting to see the transition to the strong-field regime for experimental data in which the intensity is varied and the observed effect in the angle-resolved phase for SB16(5p) can be "switched on" by a certain threshold intensity.

We are thankful for the kind comments from the referee.

We performed a new experimental measurement with a higher NIR intensity, $I_{\text{NIR}} = 3.2 \text{ TW/cm}^2$. The comparison between the emission-angle-resolved relative phase shifts of SB16 at $I_{\text{NIR}} = 1.1 \text{ TW/cm}^2$ and 3.2 TW/cm^2 is shown in Fig. R6. As the intensity of the NIR field increases, a clear variation in the angular structure is observed, implying the contribution of higher-order partial waves. The emission-angle-resolved relative phase shifts at $I_{\text{NIR}} = 3.2 \text{ TW/cm}^2$ have been added on page 7 in the supplementary information.

Also, we have updated supplementary Fig. 2 in the SI to include the β parameters extracted from the experimental RABBITT spectra with a NIR intensity of 3.2 TW/cm^2 (Shown here in Figure R7). Compared to the result at $I_{\text{NIR}} = 1.1 \text{ TW/cm}^2$ (Fig. R7 d), β_8 has a greater contribution at this higher NIR intensity, indicating an increase in g_0 - g_0 and d_0 - i_0 interference. A planned follow-up manuscript contains a more detailed exploration of the NIR intensity dependence of partial wave contributions (via RMT simulation).

To clarify this confusion, we added the following sentence in the supplementary information on page 6:

“Further evidence of the relationship between the contribution of higher-order partial waves and the RABBITT phase is given from experiment; comparing the magnitude of oscillations of the β_8 parameter (relative to the other β_n parameters) in Supplementary Figs. 2(d) and (e), it is clear that the higher-order partial waves play an increasingly important role as the NIR intensity increases. The angular distributions of $2\omega_{\text{NIR}}$ oscillation phase shifts of SB16 and SB18 with a NIR intensity of $3.2 \times 10^{12} \text{ W/cm}^2$ are shown in Supplementary Figs. 3 (a) and (b).”

Figure R6. The emission-angle-resolved phase shift distributions of the SB16(5p). **a** with a NIR intensity of $1.0 \times 10^{12} \text{ W/cm}^2$. **b** with a NIR intensity of $3.2 \times 10^{12} \text{ W/cm}^2$.

Figure R7. An updated version of Supplementary Fig. 2, to include the delay-resolved anisotropy coefficients extracted from RABBITT scans from the experiment at $I_{\text{NIR}} = 3.2 \text{ TW/cm}^2$.

We hope that we have clarified the referee's concerns in the new manuscript and that our new manuscript is suitable for publication in *Nature Communications*.

The manuscript "Heterodyne analysis of high-order partial waves in attosecond photoionization of Helium" presents measurements and simulations of an angle-resolved RABBITT setup in helium, showing that contributions of higher-order transitions (i.e., beyond two photons) can be observed in the higher-order anisotropy parameters of the photoelectron angular distribution. Particular emphasis is placed on the fact that the anisotropy parameters arise due to the interference of different partial waves, such that they can be more sensitive to the (small) amplitudes of higher partial waves than their associated probabilities would be. While the results and analysis seem correct, these physical concepts are well-known and understood, and the manuscript does not show whether this approach gives any new information on the underlying physics of the process. While the work is thus a proof-of-concept demonstration of the idea, it does not go beyond this at all, and the possible impact or potential of the method remains unclear. I thus believe that a more specialized journal such as Physical Review A would be more appropriate for this work.

We thank the referee for their review and comments on our manuscript. We believe that our results presented here are novel and insightful enough for publication in Nature Communications as the other two referees have recognized, an outlook that we hope will also be shared by both this referee and you. We would like to emphasize the significance of our work in the following.

The work is important because it directly contributes to the growing demand for methods to precisely and reliably detect few-photon effects, which underpins many techniques in ultrafast physics. Accurate interpretation of phase shifts (or, equivalently, time delays) from RABBITT measurements is required to quantify exact transition dipole matrix elements, and the contributions of individual pathways. While the referee states that many of the concepts we cover are well known, we would point out that several recent publications overlook them. A Boyer et al. in 2024, (DOI: /10.1021/acs.jpca.3c06533) employed a 2.7 TW/cm² NIR field in an angle-resolved RABBITT investigation of acetylene. In their analysis they calculate anisotropy parameters, fitting only β_0 , β_2 , β_4 . In our manuscript, we demonstrate that at even 1.0 TW/cm² higher-order partial wave contributions cannot be neglected and β_6 contributes to the 2ω signal at a similar level. Similarly, in Mayer et al. in 2020 (DOI: 10.1088/1361-6455/ab9495), the contribution of d_0 - i_0 interference to the β_8 parameter is neglected, with the authors stating that it comprises only g_0 - g_0 interference.

Further, we have clearly shown that even a small contribution from 4-photon transition pathways modulates the angular structure of the 2ω phase shift distributions, especially in the case where bound-continuum transitions via resonant states are involved. This offers new insight into the underlying physics, as this is (to our knowledge) the first evidence that higher-order transition pathways may play a significant role in the formation of 'standard' 2ω RABBITT phases. This insight is critical for the continuation of RABBITT-like investigations, as recently focus has been placed on both angle-resolved RABBITT (often necessitating the stronger NIR fields we have demonstrated to bring higher-order transitions into play) and under-threshold RABBITT (often involving transitions via bound resonant states, which we show may promote higher-order transitions).

As the referee can find in this response letter, we have further modified the manuscript following the suggestions from all referees. Thus, we believe that this improved manuscript provides a fascinating opportunity to probe ultrafast electron dynamics with a general and broad interest for the readership of *Nature Communications*.

For resubmission to any journal, the following more specific points should be addressed:

-) Overall, the manuscript is not particularly well written. There are seemingly contradictory statements, e.g., in the introduction "... at higher intensities, higher-order partial waves are unavoidable through multi-photon transitions. In this case, the contribution of higher-order partial waves should be dominated by those arising from one or two-photon transitions, making their detection impossible." - so are they unavoidable or impossible to detect? The manuscript should be carefully proofread and revised for clarity.

We thank the referee for this insight. What we intended to communicate was that the *presence* of higher-order partial waves becomes unavoidable at higher intensities, but that at present there is no mechanism to unambiguously detect their *individual* contributions.

We have modified the phrasing of this statement in the second paragraph on page 2 for clarity, changing "In this case, the contribution of higher-order partial waves should be dominated by those arising from one or two-photon transitions, making their detection impossible." to "In these cases the overall signal remains dominated by these one- or two-photon transitions, making it extremely difficult to isolate the contributions of the higher-order partial waves." Meanwhile, we carefully improved the writing of the whole manuscript which we believe is more readable.

-) The schematic in Fig. 1 is quite unclear: The thick black line is labeled both as $1s^2$ and as I_p . The "grey shadow" is almost invisible. The meaning of the blue and green regions and the diffuse lines and short markers inside them is not explained.

We thank the referee for this comment. Figure 1 has been modified as shown in Fig. R3 in our reply to referee #1.

-) The theory results show a clear double peak for SB16, while nothing of the like is visible in the experimental data. This discrepancy should be discussed.

We thank the referee for their careful review. In experimental measurements, the kinetic energy resolution is not as high as the theoretical simulations. The bandwidth of the fundamental laser also widens the frequency of high-harmonic combs, thus the photoelectrons of SB16(4p) and SB16(5p) overlap in the experimental results. To clarify this, we have modified Fig. 2 on page 2 as shown in Fig. R1 (kn response to referee #1).

-) It is claimed that when there are several contributions at the same oscillation frequency to a given sideband, "the total phase is the average of these component phases weighted by the amplitudes of the component signals". This is not correct. The sum of two sines with different phases and weights does not give a sine with the weighted average of those phases.

This has been updated in the manuscript to clarify that the total phase is well-approximated

(accurate to second order in phase difference) as the weighted average of these component phases, and a derivation supporting this has been added to the supplementary information.

The sentence “the total phase is the average of these component phases weighted by the amplitudes of the component signals” in the first paragraph on page 7 has been rephrased to “Up to second order in the phase-difference, the total phase (parameter $\varphi_{SB}^{4\omega}$) is given by the weighted average of the component phases, and the total yield (parameter B) reflects the sum of the component yields (see Supplementary Information)”

-) The explanation that the product of spherical harmonics can be expressed in the basis of spherical harmonics is well-known undergraduate-level material and its discussion could be significantly shortened. It is similarly well-known that the angular distributions depend on the interference between partial waves.

This explanation has been shortened in the manuscript and modified to clarify the importance of the d_0 - i_0 interference, although we maintain an emphasis on this mathematical background as it is critical to our explanation for why individual partial wave contributions cannot be characterized unambiguously, and establishes the link between partial wave contributions and anisotropy parameters upon which our heterodyne detection method depends.

β_8 comprises both g_0 - g_0 and d_0 - i_0 interferences, which contribute at similar magnitudes, making it impossible to use $\{\beta_{0-8}\}$ to fully characterize the contributions of the s , d_0 and g_0 waves. However, β_6 (while also being only accessible through high-order partial waves) is dominated by d_0 - g_0 interference, allowing us to clearly demonstrate the presence and effect of the g_0 waves despite being unable to fully characterize their contribution.

We agree that the mathematical background could be classified as well-known undergraduate material, but we believe it to be relevant and important enough to highlight it in the manuscript and is suitable for the broad readership for Nature Communications. To provide an example of why we believe it to be important to highlight this background, we point to the recent Mayer et al. paper (J Phys B 53, 164003 (2020)) which Referee #1 requested us to discuss. In this paper, they overlook the contribution of d_0 - i_0 interference to the β_8 parameter - stating that it comprises only g_0 - g_0 interference.

The relevant statements have been added in the first paragraph on page 9:

“Partial waves beyond the d-wave appear through β_n parameters with $n > 4$. The β_6 parameter will be characterized predominantly by interference between d_0 and g_0 waves. However, β_8 will contain contributions arising from g_0 - g_0 as well as interference d_0 - i_0 at the same order of magnitude. This prevents use of the β parameters to unambiguously characterize individual partial waves.”

We hope that we have clarified the referee's concerns in the revised manuscript and that our new manuscript is suitable for publication in *Nature Communications*.

We thank the referees for reviewing our manuscript and for their thoughtful comments and suggestions. We have revised the manuscript point by point according to their recommendations. Below, we reproduce the referee comments in black, and provide our answers in blue.

Report of Referee #1 -- NCOMMS-24-32814A

The authors have satisfactorily answered all of the referees' questions. I recommend that the present manuscript be accepted for publication.

We thank the referee for kindly recommending the publication of our work in *Nature Communications*.

Report of Referee #2 -- NCOMMS-24-32814A

The authors undertook some effort to respond to all points raised by the referees. My comments were addressed diligently and the revision of the text and the figures leads to more clarity for the reader. Additional experimental results were included in the Supporting Information, which will also help the interested reader to dive deeper into the interesting results presented in this paper. The study shows interesting new aspects on RABBITT experiments with stronger NIR fields and also lets the community reconsider older published experiments where the influence of higher order partial waves was neglected. Therefore, this paper can be considered impactful enough to justify a publication in *Nature Communications*.

We thank the referee for his/her kind summary on our modifications and the recommendation to publish our manuscript in *Nature Communications*.

Report of Referee #3 -- NCOMMS-24-32814A

The authors have carefully answered all points raised by the referees, and have significantly improved the manuscript. While I am still not fully convinced that this work rises to the level of *Nature Communications*, I think it is certainly a nice contribution and I am happy to join the other two referees in recommending publication.

We appreciate the referee's kindness and his/her recommendation to publish our manuscript in *Nature Communications*.